# LEARNING COVERAGE PATHS IN UNKNOWN ENVIRONMENTS WITH REINFORCEMENT LEARNING

## ABSTRACT

Coverage path planning (CPP) is the problem of finding a path that covers the entire free space of a confined area, with applications ranging from robotic lawn mowing and vacuum cleaning, to demining and search-and-rescue tasks. While offline methods can find provably complete, and in some cases optimal, paths for known environments, their value is limited in online scenarios where the environment is not known beforehand. In this case, the path needs to be planned online while mapping the environment. We investigate how suitable reinforcement learning is for this challenging problem, and analyze the involved components required to efficiently learn coverage paths, such as action space, input feature representation, neural network architecture, and reward function. Compared to existing classical methods, this approach allows for a flexible path space, and enables the agent to adapt to specific environment dynamics. In addition to local sensory inputs for acting on short-term obstacle detections, we propose to use egocentric maps in multiple scales based on frontiers. This allows the agent to plan a long-term path in large-scale environments with feasible computational and memory complexity. Furthermore, we propose a novel total variation reward term for guiding the agent not to leave small holes of non-covered free space. To validate the effectiveness of our approach, we perform extensive experiments in simulation with a 2D ranging sensor on different variations of the CPP problem, surpassing the performance of both previous RL-based approaches and highly specialized methods. Our code implementation can be found in the supplementary material.

## 1 INTRODUCTION

Task automation is an ever growing field, and as automated systems become more intelligent, they are expected to solve increasingly challenging problems. One such problem is coverage path planning (CPP), where a path is sought that covers the entire free space of a confined area. It is related to the traveling salesman and covering salesman problems (Galceran & Carreras, 2013), and known to be NP-hard (Arkin et al., 2000). If the environment is known *a priori*, including the extent of the area and the location and geometry of all obstacles, an optimal path can be planned *offline* (Huang, 2001). However, if the environment is initially unknown, or at least partially unknown, the path has to be planned *online*, *e.g.* by a robot, using sensors to simultaneously map the environment. In this case, an optimal path cannot generally be found (Galceran & Carreras, 2013).

Coverage path planning is an essential part in a wide variety of robotic applications, where the range of settings define different variants of the CPP problem. In applications such as robotic lawn mowing (Cao et al., 1988), snow removal (Choset, 2001), and vacuum cleaning (Yasutomi et al., 1988), the region that is covered is defined by an attached work tool. In contrast, in applications such as search-and-rescue (Jia et al., 2016), autonomous underwater exploration (Hert et al., 1996), and aerial terrain coverage (Xu et al., 2011), the covered region is defined by the range of a sensor. Additionally, the space of valid paths is limited, induced by the motion constraints of the robot.

We aim to develop a task- and robot-agnostic method for online CPP, where the environment is not known beforehand. This is a challenging task, as intricacies in certain applications may be difficult to model, or even impossible in some cases. In the real world, unforeseen discrepancies between the modeled and true environment dynamics may occur, *e.g.* due to a damaged sensor or actuator. A further aspect that needs to be considered is the sensing capability of the agent, *e.g.* if it

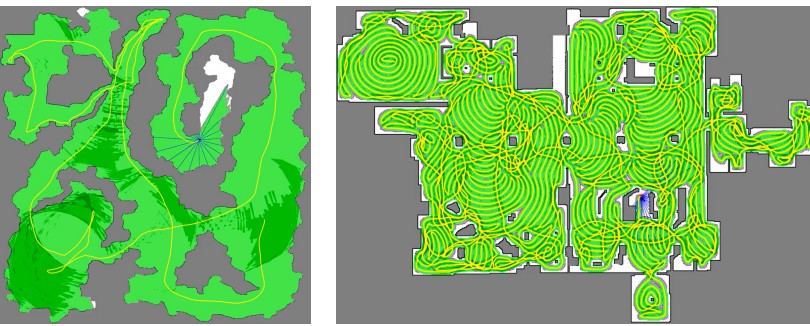

Figure 1: Examples of learned paths (yellow) for exploration (left) and lawn mowing (right).

is not omnidirectional, the agent needs to rotate itself to explore a different view of the environment. Incompleteness of the present model must be assumed to account for open world scenarios. Thus, we approach the problem from the perspective of learning through embodiment. Reinforcement learning (RL) lends itself as a natural choice, as it allows an agent to interact with the world and adapt to specific environment conditions, without the need for an explicit model. Concretely, we consider the case where an RL model directly predicts the control signals for an agent from sensor data. Besides the benefit of learning coverage paths for specific agent and environment dynamics, this approach does not constrain the path space in terms of subdivisions with pre-defined patterns as in cellular decomposition methods (Acar et al., 2002; Choset & Pignon, 1998), or primitive movements such as 90° turns and moving straight in grid-based methods (Gabriely & Rimon, 2002; Gonzalez et al., 2005). In other words, it allows for flexibility in the path space, as can be seen in Figure 1.

We analyze the different *components* in RL to efficiently learn coverage paths. First, to enable learning every valid path, a *discrete action space* used in some previous RL-based approaches (Kyaw et al., 2020; Piardi et al., 2019) is insufficient, as it excludes a large set of paths. Thus, a *continuous action space* is necessary, where the model predicts control signals for an agent. Second, to decide which action to perform, the agent needs information about its pose and the currently known state of the environment. Following Chen et al. (2019a), we use egocentric maps of the coverage and the known environment geometry. This provides a suitable *observation space*, as the egocentric maps efficiently encode the pose in an easily digestible manner. For a scalable input representation for the RL model, we propose to use multiple maps in different scales, with lower resolutions for the larger scales. With this approach, a square of size $n \times n$ can be represented in $\mathcal{O}(\log n)$ memory complexity, compared to $\mathcal{O}(n^2)$ for a single fixed-resolution map. We make this multi-scale approach viable through frontier maps that can preserve information about the existence of non-covered space at any scale. Third, a convolutional *network architecture* is a natural choice as it efficiently exploits the spatial correlation in the maps. We propose to group the maps by scale and convolve them separately, as their spatial positions do not correspond to the same location. Fourth, besides intuitive positive *reward terms* based on covering new ground (Chaplot et al., 2020; Chen et al., 2019a), we propose a novel reward term based on total variation, which guides the agent not to leave small holes of non-covered free space. Our contributions are summarized as follows:

- We propose an end-to-end reinforcement learning method for the continuous online CPP problem.
- Through frontier maps, our multi-scale map approach can represent non-visited space in any scale.
- We introduce a novel total variation reward term for eliminating islands of non-covered space.
- Finally, we conduct extensive experiments to validate the effectiveness of our approach.

## 2 RELATED WORK

Coverage path planning has previously been approached from the perspective of classical robotics, as well as reinforcement learning. Both approaches require a suitable map representation of the environment. We summarize the related works within these topics.

**Classical methods.** *Cellular decomposition methods*, such as boustrophedon cellular decomposition (BCD) (Choset & Pignon, 1998) and Morse decomposition (Acar et al., 2002), subdivide the area into several non-overlapping cells free of obstacles. Each cell is covered with a pre-defined motion, *e.g.* a back-and-forth motion. The cells are then connected using transport paths, which inevitably

lead to overlap with the already covered cells. *Grid-based methods*, such as Spiral-STC (Gabriely & Rimon, 2002) and the backtracking spiral algorithm (BSA) (Gonzalez et al., 2005), decompose the environment into uniform grid cells, such as rectangles (Zelinsky et al., 1993) or triangles (Oh et al., 2004), with a comparable size to the agent. The coverage path is defined by primitive motions between adjacent cells on the grid. The simplicity of these methods is attractive, but they heavily constrain the space of available paths. *Frontier-based methods* keep track of the boundary between covered and non-covered free space, *i.e. the frontier*. Segments of the frontier are clustered into frontier nodes, where one node is chosen as the next short-term goal. The criteria for choosing which node to visit next is based on different attributes, such as the shortest distance to the agent (Yamauchi, 1997), the path in a rapidly exploring random tree (RRT) (Umari & Mukhopadhyay, 2017), or the gradient in a potential field (Yu et al., 2021). These methods are mainly tailored for robot exploration, where the notion of a frontier is suitable. Compared to classical approaches, RL can be more robust against non-modeled environment dynamics (Saha et al., 2021c).

**RL-based methods.** Many learning-based methods combine classical components with reinforcement learning. For known environments, Chen et al. (2019b) present Adaptive Deep Path, where they apply RL to find the order in which to cover the cells in BCD. Discrete methods (Piardi et al., 2019; Kyaw et al., 2020; Saha et al., 2021b) utilize the simplicity of grid-based approaches, where an RL model predicts movement primitives, e.g. to move up, down, left, or right, based on a discrete action space. However, this constrains the space of possible paths, and may result in paths that are not possible to navigate by a robot in the real world, which is continuous by nature. Thus, we consider a continuous action space, which has been used successfully for the related navigation problem (Tai et al., 2017). Other works combine RL with a frontier-based approach, either by using an RL model to predict the cost of each frontier node (Niroui et al., 2019), or to predict continuous control signals for navigating to a chosen node (Hu et al., 2020). Different from previous approaches, which use RL within a multi-stage framework, we predict continuous control signals end-to-end to fully utilize the flexibility of RL. To the best of our knowledge, we are the first to do so for CPP.

**Map representation.** To perform coverage path planning, the environment needs to be represented in a suitable manner. Similar to previous work, we discretize the map into a 2D grid with sufficiently high resolution to accurately represent the environment. This map-based approach presents different choices for the input feature representation. Saha et al. (2021a) observe such maps in full resolution, where the effort for a $d \times d$ grid is $\mathcal{O}(d^2)$, which is infeasible for large-scale environments. Niroui et al. (2019) resize the maps to keep the input size fixed. However, this means that the information in each grid cell varies between environments, and hinders learning and generalization for differently sized environments. Instead of considering the whole environment at once, other works (Heydari et al., 2021; Saha et al., 2021b) observe a local neighborhood around the robot. While the computational cost is manageable for small neighborhoods, the long-term planning potential is limited as the space beyond the local neighborhood cannot be observed. For example, if the local neighborhood is fully covered, the robot must pick a direction at random to explore further. To avoid the limitations of these representations, we use multiple maps in different scales, similar to Klamt & Behnke (2018). Finally, Shrestha et al. (2019) propose to learn the map in unexplored regions.

## 3 LEARNING COVERAGE PATHS

Before presenting our approach for learning coverage paths, we first define the online CPP problem in Section 3.1, after which we formulate it as a Markov decision process in Section 3.2. Subsequently, we present our RL-based approach in terms of observation space in Section 3.3, action space in Section 3.4, reward function in Section 3.5, and agent architecture in Section 3.6.

### 3.1 PROBLEM DEFINITION AND LIMITATIONS

Without prior knowledge about the geometry of a confined area, the goal is to navigate in it with an agent with radius $r$ to visit every point of the free space. The free space includes all points inside the area that are not occupied by obstacles. A point is considered visited when the distance between the point and the agent is less than the coverage radius $d$, and the point is within the field-of-view of the agent. This definition unifies variations where $d \leq r$, which we define as the *lawn mowing problem*, and where $d > r$, which we refer to as *exploration*. To interactively gain knowledge about the environment, and for mapping its geometry, the agent needs some form of sensor. Both ranging

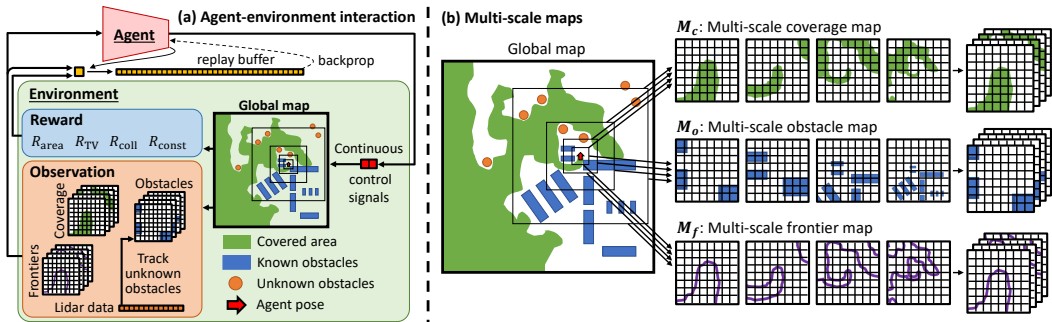

Figure 2: **(a) Agent-environment interaction:** The observation consists of multi-scale maps from (b) and lidar detections, based on which the model predicts continuous control signals for an agent. **(b) Illustration of coverage, obstacle, and frontier maps in multiple scales:** This example shows $m = 4$ scales with a scale factor of $s = 2$. All scales are centered at the agent, and discretized into the same pixel resolution, resulting in the multi-scale maps $M_c$, $M_o$, and $M_f$, of size $8 \times 8 \times 4$ here.

sensors, *e.g.* a light detection and ranging (lidar) sensor (Hu et al., 2020), and depth cameras (Chen et al., 2019a) have been utilized for this purpose. In this work, we choose a simulated 2D lidar sensor that can detect obstacles in fixed angles relative to the agent, although our proposed framework is not limited to this choice. Based on the pose of the agent, the detections are transformed to global coordinates, and a map of the environment is continuously formed. As the focus of this paper is to learn coverage paths, and not to solve the localization problem, we assume known localization up to a certain noise level. However, our method may be extended to the case with unknown pose with the use of an off-the-shelf SLAM method. Finally, while our method can be further extended to account for dynamic obstacles and multi-agent coverage, they are beyond the scope of this paper.

## 3.2 COVERAGE PATH PLANNING AS A MARKOV DECISION PROCESS

We formulate the CPP problem as a partially observable Markov decision process (POMDP) with discrete time steps $t \in [1..T]$, continuous actions $a_t \sim \pi(a_t|o_t)$ made by the agent according to policy $\pi$, observations and rewards $o_t, r_t \sim p(o_t, r_t|s_t, a_{t-1})$, and states $s_t \sim p(s_t|s_{t-1}, a_{t-1})$ based on environment dynamics. We use a neural network for the policy, and the goal for the agent is to maximize the expected discounted reward $\mathbb{E}(\sum_{t=1}^{T} \gamma^t r_t)$ with discount factor $\gamma$. In our case, the state $s_t$ consists of the geometry of the area, location and shape of all obstacles, visited points, and the pose of the agent. The observation $o_t$ contains the area and obstacle geometries mapped until time step $t$, as well as visited points, agent pose, and sensor data. Based on the observation, the model predicts control signals for the agent. The reinforcement learning loop is depicted in Figure 2(a), where observations, actions, and rewards are collected into a replay buffer, from which training batches are drawn for gradient backpropagation.

## 3.3 OBSERVATION SPACE

To efficiently learn coverage paths, the observation space needs to be mapped into a suitable input feature representation for the neural network policy. To this end, we represent the visited points as a *coverage map*, and the mapped obstacles and boundary of the area as an *obstacle map*. The maps are discretized into a 2D grid with sufficiently high resolution to accurately represent the environment. To represent large regions in a scalable manner, we propose to use multi-scale maps, which was necessary for large environments, see Appendix A.1. We make this viable through *frontier maps*, as they can preserve information about the existence of non-covered space, even in coarse scales.

**Multi-scale maps.** Inspired by multi-layered maps with varying scale and coarseness levels for search-and-rescue (Klamt & Behnke, 2018), we propose to use multi-scale maps for the coverage and obstacles to solve the scalability issue. We extract multiple local neighborhoods with increasing size and decreasing resolution, keeping the grid size fixed. We start with a local square crop $M^{(1)}$ with side length $d_1$ for the smallest and finest scale. The multi-scale map representation $M = \{M^{(i)}\}_{i=1}^{m}$ with $m$ scales is constructed by cropping increasingly larger areas based on a fixed scale

factor $s$. Concretely, the side length of map $M^{(i)}$ is $d_i = sd_{i-1}$. The resolution for the finest scale is chosen sufficiently high such that the desired level of detail is attained in the nearest vicinity of the agent, allowing it to perform precise local navigation. At the same time, the large-scale maps allow the agent to perform long-term planning, where a high resolution is less important. This multi-scale map representation can completely contain an area of size $d \times d$ in $\mathcal{O}(\log d)$ number of scales. The total number of grid cells is $\mathcal{O}(wh \log d)$, where $w$ and $h$ are the fixed width and height of the grids, and do not depend on $d$. This is a significant improvement over a single fixed-resolution map with $\mathcal{O}(d^2)$ grid cells. For the observation space, we use a multi-scale map $M_c$ for the coverage and $M_o$ for the obstacles. These are illustrated in Figure 2(b).

**Frontier maps.** When the closest vicinity is covered, the agent needs to make a decision where to explore next. However, the information in the low-resolution large-scale maps may be insufficient. For example, consider an obstacle-cluttered region where the obstacle boundaries have been mapped. A low coverage could either mean that some parts are non-covered free space, or that they are occupied by obstacles. These cases cannot be distinguished if the resolution is too low. As a solution to this problem, we propose to encode a multi-scale frontier map $M_f$, which we define in the following way. In the finest scale, a grid cell that has not been visited is a frontier point if any of its neighbours have been visited. Thus, a frontier point is non-visited free space that is reachable from the covered area. A region where the entire free space has been visited does not induce any frontier points. In the coarser scales, a grid cell is a frontier cell if and only if it contains a frontier point. In this way, the existence of frontier points persists through scales. Thus, regions with non-covered free space can be deduced in any scale, based on this multi-scale frontier map representation.

**Egocentric maps.** As the movement is made relative to the agent, its pose needs to be related to the map of the environment. Following Chen et al. (2019a), we use egocentric maps which encode the pose by representing the maps in the coordinate frame of the agent. Each multi-scale map is aligned such that the agent is in the center of the map, facing upwards. This allows the agent to easily map observations to movement actions, instead of having to learn an additional mapping from a separate feature representation of its position, such as a 2D one-hot map (Theile et al., 2020).

**Sensor observations.** To react on short-term obstacle detections, we include the sensor data in the input feature representation. The depth measurements from the lidar sensor are normalized to $[0, 1]$ based on its maximum range, and concatenated into a vector $S$ (see Appendix A.6 for implications).

## 3.4 ACTION SPACE

We let the model directly predict the control signals for the agent, which allows it to adapt to the specific agent dynamics, and overcoming the limitations of a discrete action space. We consider an agent that independently controls the linear velocity $v$ and the angular velocity $\omega$, although the action space may seamlessly be changed to specific vehicle models. To keep a fixed output range, the actions are normalized to $[-1, 1]$ based on the maximum linear and angular velocities. The sign of the linear and angular velocities controls the direction of travel and rotation respectively.

## 3.5 REWARD FUNCTION

As the goal is to cover the free space, a reward based on covering new ground is required. Similar to Chaplot et al. (2020) and Chen et al. (2019a), we define a reward term $R_{\text{area}}$ based on the newly covered area $A_{\text{new}}$ in the current time step that was not covered previously. To control the scale of this reward signal, we first normalize it to $[0, 1]$ based on the maximum area that can be covered in each time step, which is related to the maximum speed $v_{\text{max}}$. We subsequently multiply it with the scale factor $\lambda_{\text{area}}$, which reflects the highest possible reward for this term, resulting in the reward

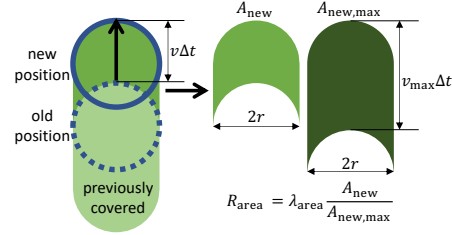

Figure 3: The area reward $R_{\text{area}}$ is based on the maximum possible area that can be covered in each time step.

$$R_{\text{area}} = \lambda_{\text{area}} \frac{A_{\text{new}}}{2rv_{\text{max}}\Delta t}, \qquad (1)$$

where $r$ is the agent radius and $\Delta t$ is the time step size. See Figure 3 for an illustration.

By only maximizing the coverage reward in (1), the agent is discouraged from overlapping its previous path, as this reduces the reward in the short term. This might lead to holes or stripes in the coverage, which we observed in our experiments (see Appendix A.4), where the agent would leave a small gap when driving adjacent to a previously covered part. These leftover parts can be costly to cover later on for reaching complete coverage. Covering the thin stripes afterward only yields a minor reward, resulting in slow convergence towards an optimal path. To reduce the leftover parts, we propose a reward term based on minimizing the total variation (TV) of the coverage map. Minimizing the TV corresponds to reducing the boundary of the coverage map, and thus leads to fewer holes. Given a 2D signal $x$, the discrete isotropic total variation, which has been used for image denoising (Rudin et al., 1992), is expressed as

$$V(x) = \sum_{i,j} \sqrt{|x_{i+1,j} - x_{i,j}|^2 + |x_{i,j+1} - x_{i,j}|^2}. \tag{2}$$

We consider two variants of the TV reward term, a global and an incremental. For the global TV reward $R_{\mathrm{TV}}^{\mathrm{G}}$, the agent is given a reward based on the global coverage map $C_t$ at time step $t$. To avoid an unbounded TV for large environments, it is scaled by the square root of the covered area $A_{\mathrm{covered}}$, as this results in a constant TV for a given shape of the coverage map, independent of scale. The incremental TV reward $R_{\mathrm{TV}}^{\mathrm{I}}$ is based on the difference in TV between the current and the previous time step. A positive reward is given if the TV is decreased, and vice versa. The incremental reward is scaled by the maximum possible increase in TV in a time step, which is twice the traveled distance. The global and incremental rewards are respectively given by

$$R_{\mathrm{TV}}^{\mathrm{G}}(t) = -\lambda_{\mathrm{TV}}^{\mathrm{G}} \frac{V(C_t)}{\sqrt{A_{\mathrm{covered}}}}, \qquad R_{\mathrm{TV}}^{\mathrm{I}}(t) = -\lambda_{\mathrm{TV}}^{\mathrm{I}} \frac{V(C_t) - V(C_{t-1})}{2v_{\mathrm{max}}\Delta t}, \tag{3}$$

where $\lambda_{\mathrm{TV}}^{\mathrm{G}}$ and $\lambda_{\mathrm{TV}}^{\mathrm{I}}$ are reward scaling parameters to make sure that $|R_{\mathrm{TV}}| < |R_{\mathrm{area}}|$ on average. Otherwise, the optimal behaviour is simply to stand still.

To avoid obstacle collisions, a negative reward $R_{\mathrm{coll}}$ is given each time the agent collides with an obstacle. Finally, a small constant negative reward $R_{\mathrm{const}}$ is given in each time step to encourage fast execution. Thus, our final reward function is written as

$$R = R_{\mathrm{area}} + R_{\mathrm{TV}}^{\mathrm{G}} + R_{\mathrm{TV}}^{\mathrm{I}} + R_{\mathrm{coll}} + R_{\mathrm{const}}. \tag{4}$$

During training, each episode is terminated when the agent reaches a pre-defined goal coverage, or when $\tau$ consecutive time steps have passed without the agent having covered any new space.

## 3.6 AGENT ARCHITECTURE

Due to the multi-modal nature of the observation space, we use a map feature extractor $g_m$, a sensor feature extractor $g_s$, and a fusing module $g_f$. The map and sensor features are fused, resulting in the control signal prediction

$$(v, \omega) = g_f(g_m(M_c, M_o, M_f), g_s(S)). \tag{5}$$

We consider two network architectures, a simple multilayer perceptron (MLP) as a baseline choice, and a scale-grouped convolutional neural network (SGCNN) utilizing an inductive prior on the spatial correlations in the maps, see Figure 4. For the MLP, the feature extractors, $g_m$ and $g_s$, are identity functions that simply flat-

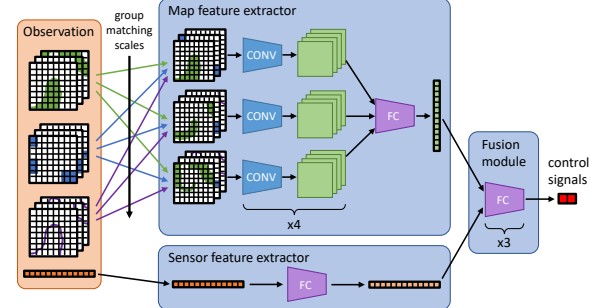

Figure 4: Our proposed agent architecture, SGCNN, consists of convolution (CONV) and fully connected (FC) layers. The scales of the multi-scale maps are convolved separately as their spatial positions are not aligned in the grid. x3/x4 refer to the number of layers.

ten the inputs, and the fusing module consists of two fully connected (FC) layers. For the SGCNN architecture, we group the maps in $M_c$, $M_o$ and $M_f$ by scale, as the pixel positions between different scales do not correspond to the same world coordinate. Each scale is convolved separately using grouped convolutions. This makes sure that grid cells are always compared to grid cells that correspond to the same spatial position. The map feature extractor additionally includes a final FC layer. The sensor feature extractor, is a single FC layer, and the fusing module consists of three FC layers. ReLU is used as the activation function throughout. Full details can be found in Appendix B.2.

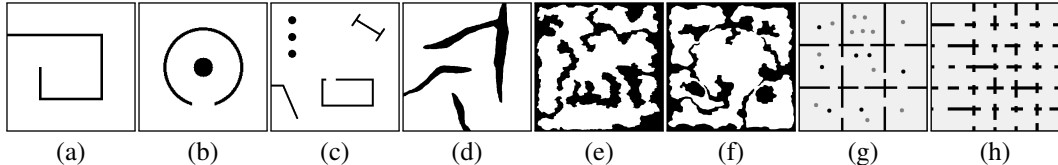

Figure 5: Examples of fixed (a-f) and randomly generated (g-h) training maps.

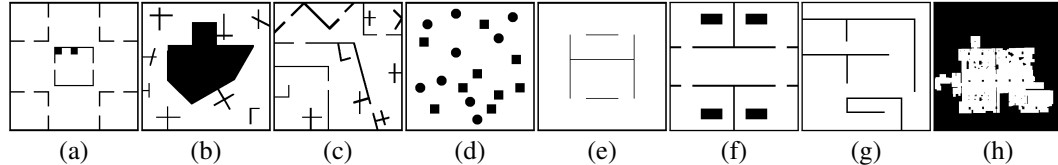

Figure 6: Examples of evaluation maps used for omnidirectional exploration (a-c), non-omnidirectional exploration (d-e), and lawn mowing (f-h).

## 4 EXPERIMENTS

In this section, we first describe the implementation details including the training of the agent and the setup of the environment in Section 4.1. Subsequently, we evaluate our method in *omnidirectional exploration* in Section 4.2, *non-omnidirectional exploration* in Section 4.3, and on the *lawn mowing* CPP problem in Section 4.4. Finally, we provide ablation studies in Section 4.5.

### 4.1 IMPLEMENTATION DETAILS

**Agent and training details.** We use soft actor-critic (SAC) RL (Haarnoja et al., 2018a) with automatic temperature tuning (Haarnoja et al., 2018b). The actor and critic networks share the network architecture, but not any weights. We train for 4M iterations with learning rate $5 \cdot 10^{-5}$, batch size 256, replay buffer size $5 \cdot 10^5$, and discount factor $\gamma = 0.99$. The simulated lidar field-of-view was $360°$ in omnidirectional exploration, and $180°$ in the other two settings. The coverage radius was set to the lidar range in both exploration tasks, which was 7 m and 3.5 m in omnidirectional and non-omnidirectional exploration respectively. In the lawn mowing case, the coverage radius was the agent radius, which was 0.15 m. For full details on the physical dimensions, see Appendix B.1. The training time for one agent varied between 50 and 100 hours on a T4 GPU and a 6226R CPU.

**Environment details.** For the multi-scale maps, we use $m = 4$ scales with $32 \times 32$ pixel resolution, a scale factor of $s = 4$, and 0.0375 meters per pixel for the finest scale, which corresponds to 4 pixels per robot radius. Thus, the maps span a square with side length 76.8 m. We progressively increase the difficulty of the environment during training, starting from simple maps and increase their complexity over time. The episodes are terminated when a goal coverage rate between $90-99\%$ is reached, depending on the training progression. For full details on the progressive training, see Appendix B.5. Based on initial experiments, we find suitable reward parameters. The episodes are prematurely truncated if $\tau = 1000$ consecutive steps have passed without the agent covering any new space. We set the maximum coverage reward $\lambda_{\text{area}} = 1$, the incremental TV reward scale $\lambda_{\text{TV}}^{\text{I}} = 0.2$ for exploration and $\lambda_{\text{TV}}^{\text{I}} = 1$ for lawn mowing, the collision reward $R_{\text{coll}} = -10$, and the constant reward $R_{\text{const}} = -0.1$. The global TV reward scale was set to $\lambda_{\text{TV}}^{\text{G}} = 0$ as it did not contribute to a performance gain in our ablations in Section 4.5.

**Map geometry.** To increase the variation of training scenarios, we use both a fixed set of maps and randomly generated maps, see Figure 5. The fixed maps provide both simple, and more challenging scenarios that a CPP method is expected to solve. The randomly generated maps provide orders of magnitude more variation, and avoid the problem of overfitting to the fixed maps. They are created by randomizing grid-like floor plans with random obstacles. For more details, see Appendix B.4.

**Evaluation.** We measure the times $T_{90}$ and $T_{99}$ to reach 90% and 99% coverage respectively. We use separate maps for evaluation that are not seen during training. See Figure 6 for examples, and Appendix B.6 for a full list of evaluation maps used in the different CPP variations. Additionally, see Appendix A.2 for inference times, and Appendix A.3 for an evaluation on collision frequency.

### 4.2 OMNIDIRECTIONAL EXPLORATION

To evaluate our method on omnidirectional exploration, we use Explore-Bench (Xu et al., 2022), which is a recent benchmark for exploration that implements relevant baseline methods and environments. It includes six environments; loop, corridor, corner, rooms, combination 1 (rooms with corridors), and combination 2 (complex rooms with tight spaces), which can be found in Appendix B.6, Figure 11(a), ordered left to right. We compare our method against a distance-based frontier method (Yamauchi, 1997), an RRT-based frontier method (Umari & Mukhopadhyay, 2017), and a potential field-based frontier method (Yu et al., 2021). We also compare with an RL-based approach, where Xu et al. (2022) train an RL model to determine a global goal based on active neural SLAM (Chaplot et al., 2020). As the baseline methods estimate the position with SLAM, we measure their average position error to 1 cm. For a fair comparison, we add Gaussian noise to the position in our method with a corresponding standard deviation. The results are presented in Table 1, where our approach surpasses all methods in the comparison. This shows that learning control signals end-to-end with RL is in fact a suitable approach to CPP. Remarkably, this approach is even robust to noise, which highlights that the agent can adapt to uncertainties in the observed state of the environment.

Table 1: Time in seconds for reaching $90\%$ and $99\%$ coverage in Explore-Bench.

| Map → | Loop | | Corridor | | Corner | | Rooms | | Comb. 1 | | Comb. 2 | | Total | |
|---|---|---|---|---|---|---|---|---|---|---|---|---|---|---|
| Method ↓ | $T_{90}$ | $T_{99}$ | $T_{90}$ | $T_{99}$ | $T_{90}$ | $T_{99}$ | $T_{90}$ | $T_{99}$ | $T_{90}$ | $T_{99}$ | $T_{90}$ | $T_{99}$ | $T_{90}$ | $T_{99}$ |
| Distance frontier | 124 | 145 | 162 | 169 | 210 | 426 | 159 | 210 | 169 | 175 | 230 | 537 | 1054 | 1662 |
| RRT frontier | 145 | 180 | 166 | 170 | 171 | 331 | 176 | 211 | 141 | 192 | 249 | 439 | 1048 | 1523 |
| Pot. field frontier | 131 | 152 | 158 | 162 | 133 | 324 | 156 | 191 | 165 | 183 | 224 | 547 | 967 | 1559 |
| Act. Neur. SLAM | 190 | 214 | 160 | 266 | 324 | 381 | 270 | 315 | 249 | 297 | 588 | 755 | 1781 | 2228 |
| Ours (with noise) | **99** | **109** | **84** | 120 | 123 | 489 | 83 | **165** | 97 | **101** | **149** | 420 | **635** | 1404 |
| Ours (no noise) | 126 | 137 | 95 | **100** | **76** | **302** | **71** | 250 | **87** | 118 | 195 | **353** | 650 | **1260** |

### 4.3 NON-OMNIDIRECTIONAL EXPLORATION

We further evaluate our method on non-omnidirectional exploration, using the maps in Appendix B.6, Figure 11(b). We compare with a recent frontier-based RL method (Hu et al., 2020), and a random agent baseline. The random agent drives forward until it encounters an obstacle, where it turns a random amount, and repeats. The coverage over time is presented in Figure 7. Our method outperforms the frontier-based RL approach, demonstrating that end-to-end learning of control signals is superior to a multi-stage approach for adapting to a specific sensor setup.

### 4.4 LAWN MOWING

For the lawn mowing problem, where the coverage radius is the same as the agent radius, we compare with the backtracking spiral algorithm (BSA) (Gonzalez et al., 2005), which is a common benchmark for this CPP variation. Note however that BSA is an offline method and does not solve the mapping problem. As such, we do not expect our approach to outperform it in this comparison. Instead, we use it to see how close we are to a good solution. Table 3 shows $T_{90}$ and $T_{99}$ for six maps numbered 1-6, which can be found in Appendix B.6, Figure 11(c), ordered left to right. In total, our method takes $29\%$ and $49\%$ more time to reach $90\%$ and $99\%$ coverage respectively. This is an impressive result considering the challenge of simultaneously mapping the environment.

### 4.5 ABLATION STUDY

In Table 2, we explore the impact of different components of our approach via a series of ablations.

**TV rewards.** We find that the incremental TV reward has a major impact in the lawn mowing problem, increasing the coverage from $85.1\%$ to $97.8\%$ (③ vs. ⑥). It also affects exploration, but not to the same extent. This is reasonable, as the total variation in the coverage map is lower in this case. Moreover, as global TV has less temporal variation, and behaves more like a constant reward, it turns out not to be beneficial for CPP (⑤ vs. ⑥).

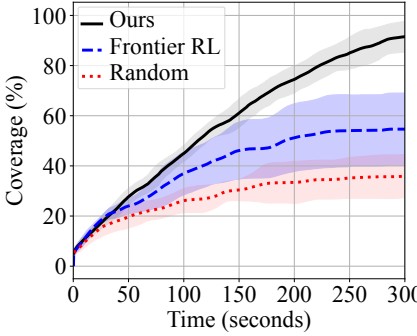

Figure 7: Coverage over time in non-omnidirectional exploration.

Table 2: Coverage (%) at 1500 and 1000 seconds for mowing (Mow) and exploration (Exp) respectively, comparing agent architecture (NN), TV rewards, and frontier observation ($F_{obs}$).

| | | Settings | | | Coverage | |
|---|---|---|---|---|---|---|
| | $R_{TV}^I$ | $R_{TV}^G$ | $F_{obs}$ | NN | Mow | Exp |
| ① | ✓ | | ✓ | MLP | 81.4 | 27.3 |
| ② | ✓ | | ✓ | CNN | 93.2 | 88.4 |
| ③ | | | ✓ | SGCNN | 85.1 | 91.5 |
| ④ | ✓ | | | SGCNN | 72.6 | 80.6 |
| ⑤ | ✓ | ✓ | ✓ | SGCNN | 96.6 | 89.0 |
| ⑥ | ✓ | | ✓ | SGCNN | **97.8** | **93.0** |

Table 3: Time in minutes for reaching 90% and 99% coverage in lawn mowing.

| Map → | Map 1 | | Map 2 | | Map 3 | | Map 4 | | Map 5 | | Map 6 | | Total | |
|---|---|---|---|---|---|---|---|---|---|---|---|---|---|---|
| Method | $T_{90}$ | $T_{99}$ | $T_{90}$ | $T_{99}$ | $T_{90}$ | $T_{99}$ | $T_{90}$ | $T_{99}$ | $T_{90}$ | $T_{99}$ | $T_{90}$ | $T_{99}$ | $T_{90}$ | $T_{99}$ |
| BSA | 30 | 35 | 29 | 35 | 31 | 36 | 34 | 41 | 17 | 23 | 88 | 100 | 229 | 270 |
| Ours | 42 | 55 | 36 | 52 | 43 | 54 | 43 | 51 | 33 | 44 | 99 | 146 | 296 | 402 |

**Agent architectures.** The higher model capacity of the CNN architecture enables a better understanding of the environment, and outperforms the MLP baseline (② vs. ①). Our proposed architecture (SGCNN, ⑥), which groups the different scales and convolves them separately, further improves the performance compared to a naive CNN, which treats all scales as the same spatial structure.

**Multi-scale frontier maps.** We find that our proposed frontier map representation is crucial for achieving a high performance with the multi-scale map approach (④ vs. ⑥). Without this input feature, too much information is lost in the coarser scales, which is needed for finding efficient coverage paths.

**Number of scales.** In Figure 8, we compare the coverage for different number of scales on the exploration task. It

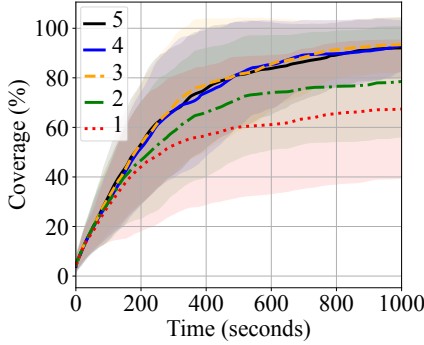

Figure 8: Exploration coverage over time for different number of scales.

shows that only using one or two scales is not sufficient, where at least three scales are required to represent a large enough environment to enable long-term planning. The discrepancy was not as large for the lawn mowing problem, as it is more local in nature with a lower coverage radius.

## 5 CONCLUSIONS

We present a method for online coverage path planning in unknown environments, based on a continuous end-to-end reinforcement learning approach. The agent implicitly solves three tasks simultaneously, namely *mapping* of the environment, *planning* a coverage path, and *navigating* the planned path while avoiding obstacles. For a scalable solution, we propose to use multi-scale maps, which is made viable through frontier maps that preserve the existence of non-covered space in coarse scales. Furthermore, we propose a novel reward term based on total variation, to reduce islands of non-covered space. This results in a task- and robot-agnostic CPP method that can be applied to different applications, which has been demonstrated for the exploration and lawn mowing tasks.

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

APPENDIX

This appendix contains an extended analysis in Section A and additional implementation details in Section B.

## A    EXTENDED ANALYSIS

### A.1    THE NECESSITY OF MULTI-SCALE MAPS

While the multi-scale maps can represent large regions efficiently, they are even necessary for large-scale environments. In large environments, using a single map is not feasible as the computational cost is $\mathcal{O}(n^2)$. For example, with our current setup with 4 maps that span an area of size $76.8 \times 76.8$ m$^2$, the training step time is 40 ms. With a single map spanning only $19.2 \times 19.2$ m$^2$ using the same pixel resolution as the finest scale, the training step time is 2500 ms. This would increase the total training time by roughly two orders of magnitude. Going beyond this size resulted in memory problems on a T4 16GB GPU. Thus, a multi-scale map is necessary for large-scale environments.

### A.2    INFERENCE TIME ANALYSIS

We evaluate the inference time of our method on two different systems in Table 4. System 1 is a high-performance computing cluster, with 16-core 6226R CPUs, and T4 GPUs. However, only 4 CPU cores and one GPU were allocated for the analysis. System 2 is a laptop computer, with a 2-core i5-520M CPU, without a GPU. The times are compared between the mowing and exploration tasks, where the inference time is lower for the mowing task since a smaller local neighborhood is used for updating the global coverage map. As our network is fairly lightweight it can run in real time, i.e. $\geq 20$ frames per second, even on a low-performance laptop without a GPU. All map updates are performed locally, resulting in high scalability, reaching real-time performance on the cluster node, and close to real-time performance on the laptop.

Table 4: Inference time in milliseconds, on a high-performance computing cluster and a laptop, for the model forward pass, other components (map updates, observation creation etc.), and in total.

| Task | Cluster node 6226R CPU, T4 GPU | | | Laptop i5-520M CPU, No GPU | | |
|---|---|---|---|---|---|---|
| | model | other | total | model | other | total |
| Exploration | 2 | 19 | 21 | 5 | 57 | 62 |
| Mowing | 2 | 3 | 5 | 5 | 9 | 14 |

### A.3    COLLISION FREQUENCY

In most real-world applications, it is vital to minimize the collision frequency, as collisions can inflict harm on humans, animals, the environment, and the robot. To gain insights into the collision characteristics of our trained agent, we tracked the collision frequency during evaluation. For exploration, it varied between once every 100-1000 seconds, which is roughly once every 50-500 meters. For lawn mowing, it varied between once every 150-250 seconds, which is roughly once every 30-50 meters. These include all forms of collisions, even low-speed side collisions. The vast majority of collisions were near-parallel, and we did not observe any head on collisions. The practical implications are very different between these cases.

### A.4    IMPACT OF THE TV REWARD ON LEARNED PATHS

Figure 9 shows learned paths both with and without the incremental TV reward. The TV reward reduces small leftover regions which are costly to cover later on.

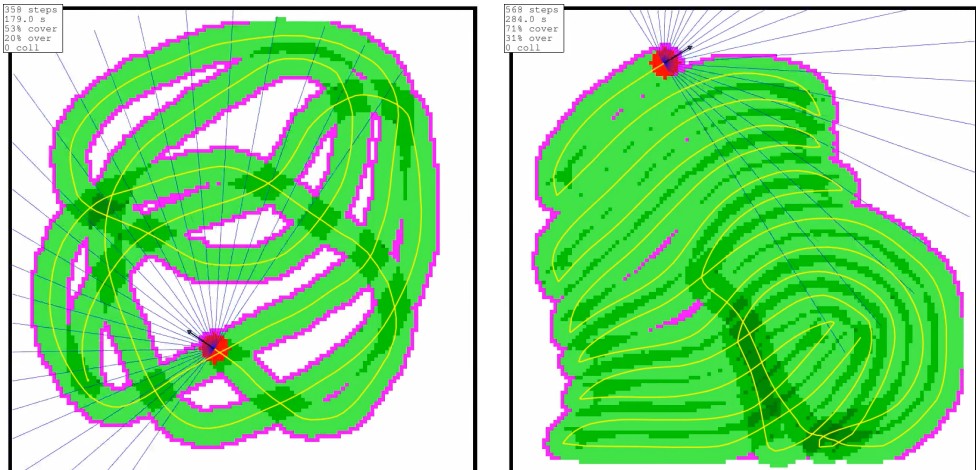

Figure 9: The total variation reward term improves path quality. The figure shows the path without TV reward (left), and with TV reward (right).

### A.5  ADDITIONAL REWARDS THAT DID NOT WORK

In addition to the reward terms described in Section 3.5, we experimented with a goal coverage reward, and an episode truncation reward. Both were added at the end of every episode. The goal coverage reward was a positive reward that was given once the agent reached the predefined goal coverage. The truncation reward was a negative reward that was given in case the episode was prematurely truncated due to the agent not covering new ground in $\tau$ consecutive steps. However, neither of these rewards affected the end result, likely due to the fact that they are sparse compared to the other rewards, and do not provide a strong reward signal. The coverage reward already encourages complete coverage, and the constant negative reward heavily penalizes not reaching full coverage. Since both of these are dense, adding sparse rewards for the same purpose does not provide any benefit.

### A.6  GENERALIZATION OF THE LIDAR FEATURE REPRESENTATION

Since we represent the lidar distance measurements as a vector of a fixed size and normalize the distances to $[0, 1]$, the sensor observation is tied to a specific sensor setup. Due to the normalization, the sensor observation is tied to a specific sensor range. This is important to keep in mind when transferring to another sensor setup. If the maximum range differs, the normalization constant needs to be considered, such that the same distance corresponds to the same observed value, and the maximum observable value should be limited to 1. If the number of lidar rays differ, the representation might not generalize as easily, as the dimensionality of the sensor vector would change. One option is to interpolate the observed distances at the ray angles that were used during training. Although, inaccuracies may occur if the angles are not well aligned. However, the map representation is more flexible, and does in theory also contain the information of the raw lidar data. So excluding the sensor feature extractor would in theory generalize to any sensor setup.

## B  ADDITIONAL IMPLEMENTATION DETAILS

### B.1  PHYSICAL DIMENSIONS

Table 5 shows the physical dimensions in the three different variations of the CPP problem that were considered. We match the dimensions of the methods under comparison to get comparable performance values.

Table 5: Physical dimensions for the three different CPP variations.

|  | Omnidirectional exploration | Non-omnidirectional exploration | Lawn mowing |
|---|---|---|---|
| Coverage radius | 7 m | 3.5 m | 0.15 m |
| Agent radius | 0.08 m | 0.15 m | 0.15 m |
| Maximum linear velocity | 0.5 m/s | 0.26 m/s | 0.26 m/s |
| Maximum angular velocity | 1 rad/s | 1 rad/s | 1 rad/s |
| Simulation step size | 0.5 s | 0.5 s | 0.5 s |
| Lidar rays | 20 | 24 | 24 |
| Lidar range | 7 m | 3.5 m | 3.5 m |
| Lidar field-of-view | 360° | 180° | 180° |

## B.2 AGENT ARCHITECTURE DETAILS

As proposed for soft actor-critic (SAC) (Haarnoja et al., 2018b), we use two Q-functions that are trained independently, together with accompanying target Q-functions whose weights are exponentially moving averages. No separate state-value network is used. All networks, including actor network, Q-functions, and target Q-functions share the same network architecture, which is either a multilayer perceptron (MLP), a naive convolutional neural network (CNN), or our proposed scale-grouped convolutional neural network (SGCNN). These are described in the following paragraphs.

**MLP.** As mentioned in the main paper, the map and sensor feature extractors for MLP simply consist of flattening layers that restructure the inputs into vectors. The flattened multi-scale coverage maps, obstacle maps, frontier maps, and lidar detections are concatenated and fed to the fusion module. For the Q-functions, the predicted action is appended to the input of the fusion module. The fusion module consists of two fully connected layers with ReLU and 256 units each. A final linear prediction head is appended, which predicts the mean and standard deviation for sampling the action in the actor network, or the Q-value in the Q-functions.

**Naive CNN.** This architecture is identical to SGCNN, see below, except that it convolves all maps simultaneously in one go, without utilizing grouped convolutions.

**SGCNN.** Our proposed SGCNN architecture uses the same fusion module as MLP, but uses additional layers for the feature extractors, including convolutional layers for the image-like maps. Due to the simple nature of the distance readings for the lidar sensor, the lidar feature extractor only uses a single fully connected layer with ReLU. It has the same number of hidden neurons as there are lidar rays. The map feature extractor consists of a $2 \times 2$ convolution with stride 2 to reduce the spatial resolution, followed by three $3 \times 3$ convolutions with stride 1 and without padding. We use 24 total channels in the convolution layers. Note that the maps are grouped by scale, and convolved separately by their own set of filters. The map features are flattened into a vector and fed to a fully connected layer of 256 units, which is the final output of the map feature extractor. ReLU is used as the activation function throughout.

## B.3 CHOICE OF HYPERPARAMETERS

**Multi-scale map parameters.** We started by considering the size and resolution in the finest scale, such that sufficiently small details could be represented in the nearest vicinity. We concluded that a resolution of 0.0375 meters per pixel was a good choice as it corresponds to 8 pixels for a robot with a diameter of 30 cm. Next, we chose a size of $32 \times 32$ pixels, as this results in the finest scale spanning $1.2 \times 1.2$ meters. Next, we chose the scale factor $s = 4$, as this allows only a few maps to represent a relatively large region, while maintaining sufficient detail in the second finest scale. Finally, the training needs to be done with a fixed number of scales. The most practical way is to simply train with sufficiently many scales to cover the maximum size for a particular use case. Thus, we chose $m = 4$ scales, which in total spans a $76.8 \times 76.8$ m region, and can contain any of the considered training and evaluation environments. If the model is deployed in a small area, the larger scales would not contain any frontier points in the far distance, but the agent can still cover the area

by utilizing the smaller scales. For a larger use case, increasing the size of the represented area by adding more scales is fairly cheap, as the computational cost is $\mathcal{O}(\log n)$.

**Reward parameters.** Due to the normalization of the area and TV rewards, they are given an equal weight with a maximum reward of one for $R_{\text{area}}$ and $R_{\text{TV}}^{\text{I}}$, with $\lambda_{\text{area}} = 1$ and $\lambda_{\text{TV}}^{\text{I}} = 1$. $R_{\text{TV}}^{\text{G}}$ is close to one for $\lambda_{\text{TV}}^{\text{G}} = 1$. Thus, we used this as a starting point, which seemed to work well for the lawn mowing problem, while a lower value was advantageous for exploration.

**Episode truncation.** For large environments, we found it important not to truncate the episodes too early, as this would hinder learning. If the truncation parameter $\tau$ was set too low, the agent would not be forced to learn to cover an area completely, as the episode would simply truncate whenever the agent could not progress, without a large penalty. With the chosen value of $\tau = 1000$, the agent would be greatly penalized by the constant reward $R_{\text{const}}$ for not reaching the goal coverage, and would be forced to learn to cover the complete area in order to maximize the reward.

### B.4 RANDOM MAP GENERATION

Inspired by procedural environment generation for reinforcement learning (Justesen et al., 2018), we use randomly generated maps during training to increase the variation in the training data and to improve generalization. We consider random floor plans and randomly placed circular obstacles. First, the side length of a square area is chosen uniformly at random, from the interval $[2.4, 7.5]$ meters for mowing and $[9.6, 15]$ meters for exploration. Subsequently, a random floor plan is created with a 70% probability. Finally, with 70% probability, a number of obstacles are placed such that they are far enough apart to guarantee that the agent can visit the entire free space, i.e. that they are not blocking the path between different parts of the free space. An empty map can be generated if neither a floor plan is created nor obstacles placed. In the following paragraphs, we describe the floor plan generation and obstacle placement in more detail.

**Random floor plans.** The random floor plans contain square rooms of equal size in a grid-like configuration, where neighboring rooms can be accessed through door openings. On occasion, a wall is removed completely or some openings are closed off to increase the variation further. First, floor plan parameters are chosen uniformly at random, such as the side length of the rooms from $[1.5, 4.8]$ meters, wall thickness from $[0.075, 0.3]$ meters, and door opening size from $[0.6, 1.2]$ meters. Subsequently, each vertical and horizontal wall spanning the whole map either top-to-bottom or left-to-right is placed with a 90% probability. After that, door openings are created at random positions between each neighboring room. Finally, one opening is closed off at random for either each top-to-bottom spanning vertical wall or each left-to-right spanning horizontal wall, not both. This is to ensure that each part of the free space can be reached by the agent.

**Random circular obstacles.** Circular obstacles with radius 0.25 meters are randomly scattered across the map, where one obstacle is placed for every four square meters. If the closest distance between a new obstacle and any previous obstacle or wall is less than 0.6 meters, the new obstacle is removed to ensure that large enough gaps are left for the agent to navigate through and that it can reach every part of the free space.

### B.5 PROGRESSIVE TRAINING

To improve convergence in the early parts of training, we apply curriculum learning (Bengio et al., 2009), which has shown to be effective in reinforcement learning (Narvekar et al., 2020). We use simple maps and increase their difficulty progressively. To this end, we rank the fixed training maps by difficulty, and assign them into tiers, see Figure 10. The maps in the lower tiers have smaller sizes and simpler obstacles. For the higher tiers, the map size is increased together with the complexity of the obstacles.

Furthermore, we define levels containing certain sets of maps and specific goal coverage percentages, see Table 6. The agent starts at level 1, and progresses through the levels during training. To progress to the next level, the agent has to complete each map in the current level, by reaching the specified goal coverage. Note that randomly generated maps are also used in the higher levels, in which case the agent has to additionally complete a map with random floor plans, and a map with randomly placed circular obstacles to progress. Whenever random maps are used in a level, either a fixed map or a random map is chosen with a 50% probability each at the start of every episode.

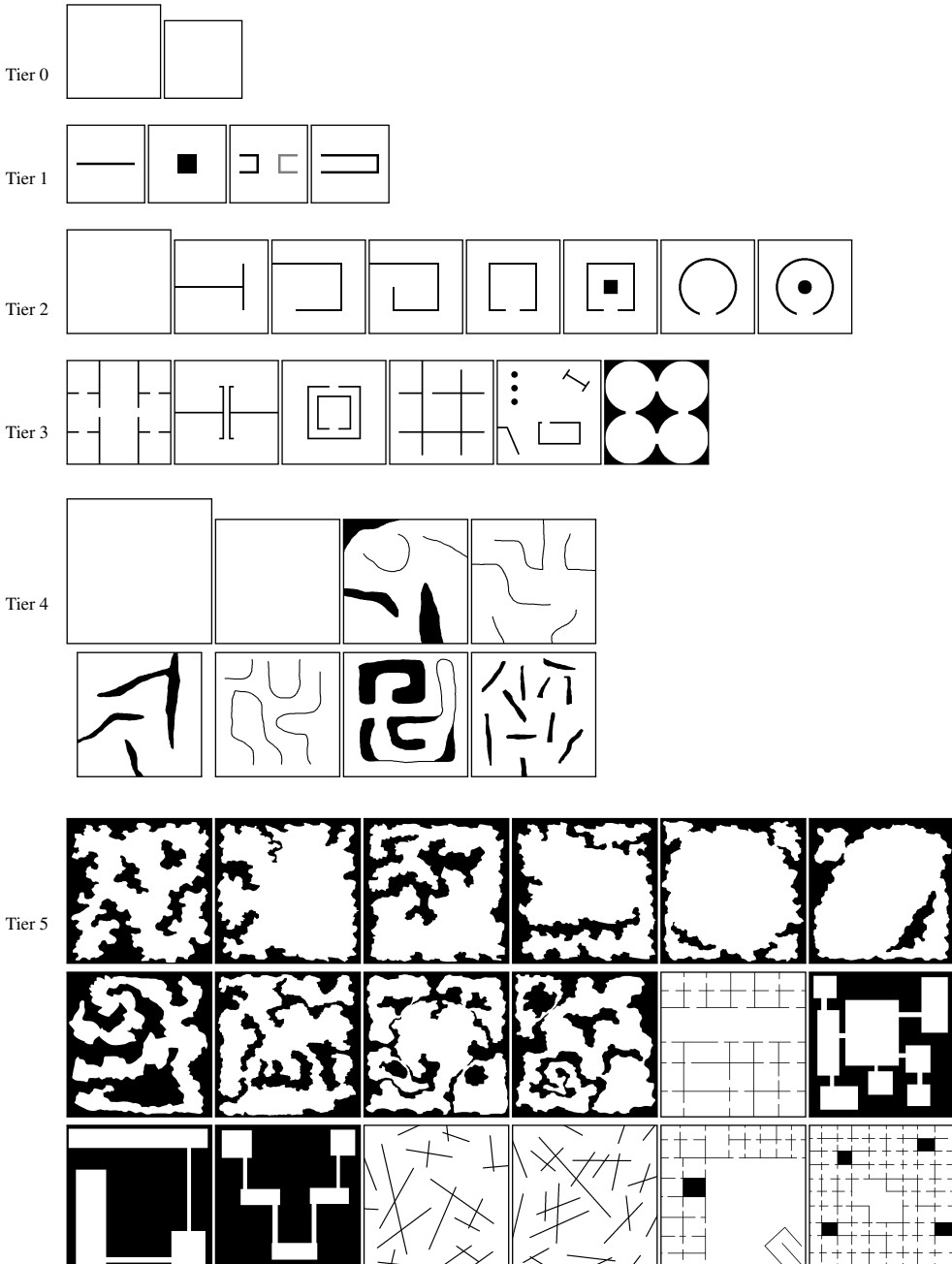

Figure 10: The fixed training maps are grouped into tiers by difficulty, depending on their size and complexity of the obstacles.

## B.6 EVALUATION MAPS

Figure 11 shows the maps used for evaluating our method on the different CPP variations, as well as additional maps for the ablation study.

## B.7 IMPLEMENTATION OF THE COMPARED METHODS

Table 7 lists the code implementations used for the methods under comparison. Note that for omnidirectional exploration, we evaluate our method under the same settings as in Explore-Bench, for

Table 6: The progressive training levels contain maps of increasingly higher tiers and goal coverage percentages. At the highest levels, generated maps with random floor plans and obstacles are also used.

| | Exploration | | | Lawn mowing | | |
|---|---|---|---|---|---|---|
| Level | Map tiers | Random maps | Goal coverage | Map tiers | Random maps | Goal coverage |
| 1 | 1,2 | | 90% | 0 | | 90% |
| 2 | 1,2,4 | | 90% | 0,1 | | 90% |
| 3 | 1,2,4 | | 95% | 0,1 | | 95% |
| 4 | 1,2,4 | | 97% | 0–2 | | 95% |
| 5 | 1,2,4 | | 99% | 0–2 | | 97% |
| 6 | 1–4 | | 99% | 0–2 | | 99% |
| 7 | 1–4 | ✓ | 99% | 0–3 | | 99% |
| 8 | 1–5 | ✓ | 99% | 0–3 | ✓ | 99% |

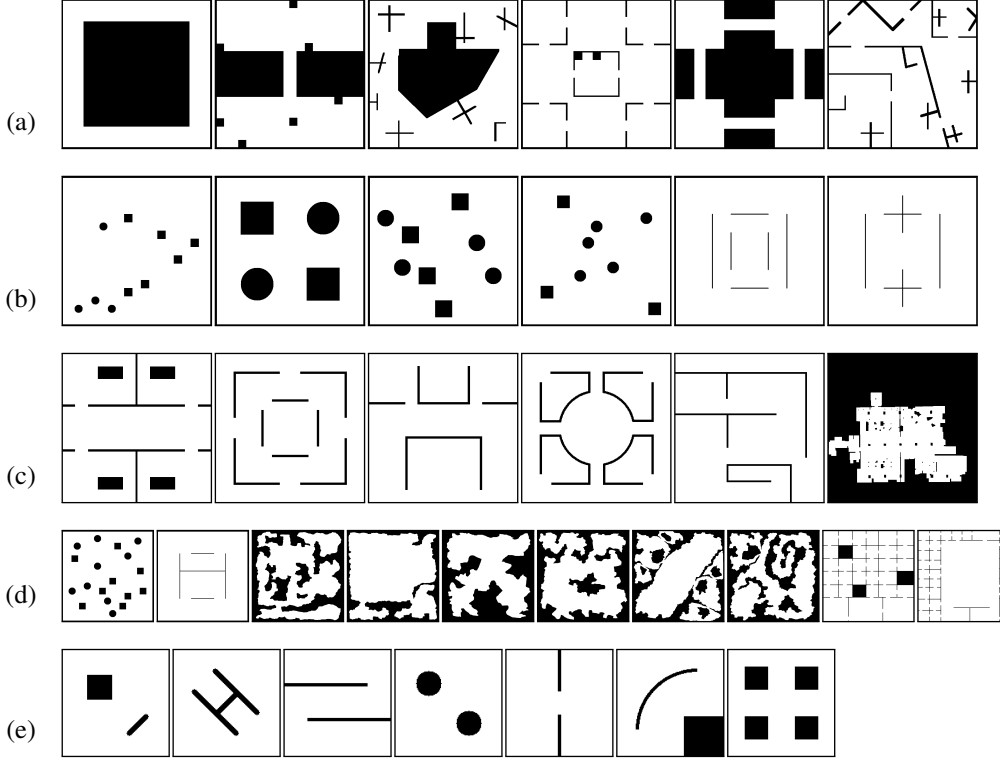

Figure 11: Evaluation maps used for (a) omnidirectional exploration, (b) non-omnidirectional exploration, and (c) lawn mowing. The maps in the last two rows were used additionally for ablations in (d) exploration and (e) lawn mowing.

which Xu et al. (2022) report the performance of the compared methods. We report these results in Table 1, and use their official implementation to estimate the position noise.

Table 7: Links to code implementations used for the methods under comparison.

---

**Omnidirectional exploration**
Official Explore-Bench implementation (Xu et al., 2022):
`https://github.com/efc-robot/Explore-Bench`

---

**Non-omnidirectional exploration**
Official Frontier RL implementation (Hu et al., 2020):
`https://github.com/hanlinniu/turtlebot3_ddpg_collision_avoidance`
Frontier RL implementation in simulation:
`https://github.com/Peace1997/Voronoi_Based_Multi_Robot_Collaborate_Exploration_Unknow_Enviroment`

---

**Lawn mowing**
Implementation of BSA (Gonzalez et al., 2005):
`https://github.com/nobleo/full_coverage_path_planner`

---

