# OpenReview forum: "Learning Coverage Paths in Unknown Environments with Reinforcement Learning"
_ICLR.cc/2024/Conference — Submitted to ICLR 2024_

### Official Review · Reviewer_5fHE · 2023-10-29

**Soundness:** 3 good
**Presentation:** 4 excellent
**Contribution:** 3 good
**Rating:** 5
**Confidence:** 4

**Summary:**

This work proposes and investigates the effectiveness of a deep RL method for coverage path planning, with lawn mowing and exploration as practical example tasks. While the ultimate objective is total area covered, the authors proposes an additional total variation objective which minimizes the perimeter of explored areas to encourage coherent exploration, in addition to other collision objectives. The observation consists of multi-scaled and ego-centric maps of areas explored as well as frontiers, while the action space is the continuous speed and angular velocity of the agent. The authors trains the agent on a set of generated maps and evaluates on similar maps and additional maps from Explore-Bench. Favorable performance is demonstrated compared to frontier-based heuristics and some RL approaches.

**Strengths:**

The proposed method seems relatively simple and intuitive from a learning perspective, which improves the chances that it might generalize beyond the paper. I have not been able to find previous works which are substantively similar, which speaks to the originality of the overall method and its application, even though individual components such as multi-scale observations, continuous actions, and reward shaping are not original by themselves. The clarity is good and the paper is easy to read. The experimental results are convincing compared to the baseline methods, and the experiments and ablations are appropriately designed.

**Weaknesses:**

My main concern is that the baselines may not be challenging enough. Intuitively, I agree on the focus on frontier-based baselines, but there should be a traveling salesman (TSP)-based baseline which aims to find a shortest path among the frontier points with a state-of-the-art TSP solver like Concorde, either in a discretized grid (converted into a graph) or in a probabilistic roadmap (PRM) graph. As mentioned in the survey below, TSP is a common technique applicable in coverage path problems.
```
Tan, Chee Sheng, Rosmiwati Mohd-Mokhtar, and Mohd Rizal Arshad. "A comprehensive review of coverage path planning in robotics using classical and heuristic algorithms." *IEEE Access* 9 (2021): 119310-119342.
```
The RL-based baseline (Hu 2020) targets a multi-agent setting and is a simple feed-forward policy with no convolutional architecture, and it’s not clear if this is a fair comparison.

Overall, there is very little detail on how the baselines are implemented in this work; many of the baselines are originally proposed in multi-agent settings rather than single-agent settings. It’s difficult to judge how non-trivial the baselines are, and intuitively there should be a TSP-based or other combinatorial optimization-based baseline, given that CPP is a combinatorial problem. While the paper presents strong results relative to the given baselines, the baselines themselves have to be appropriately designed.

Minor:
I think the total variation contribution is slightly over-claimed, as the total variation is simply the perimeter of the explored area. While this shaping term is interesting and useful, using “total variation” to describe this reward makes the paper less clear and does not provide additional insights / intuition.

**Questions:**

I see the collision penalty in the objective term. Does the agent collide at all with obstacles? If so, I would like to see the frequency of collision of the different approaches.

---

> ### Author Response · Authors · 2023-11-17
> **Replies to reviewer 5fHE**
>
> We thank the reviewer for their valuable feedback on how to improve the comparison with other methods, and we appreciate the positive feedback. We reply to the concerns below.
>
> __The baselines may not be challenging enough, a TSP-baseline should be included.__ We agree that a TSP-baseline would further increase the quality of the comparison. However, we believe that the methods under comparison provide sufficient insights about the performance of our method, as they span a wide range of approaches.
>
> __Unclear how the baselines are implemented.__ We used the implementations from their code repositories. We will provide links to these in the appendix to make it more clear.
>
> __The RL-based baseline (Hu et al., 2020) targets a multi-agent setting, it is not clear if this is a fair comparison.__ As Hu et al. predict control signals using RL (although not end-to-end) we consider it the closest related method to our work, and thus we wanted to compare with it. Based on Voronoi partitioning, it can also be used for multi-agent exploration. However, for a fair comparison, we limited it to one agent as this is the focus of our paper. As Hu et al. predict the control signals based on the relative goal position from a previous stage in their framework, a simpler network architecture is sufficient, compared to predicting them based on the full map observation.
>
> __Some baselines are originally proposed in multi-agent settings.__ Apart from Hu et al. and the potential field-based method (Yu et al., 2021), all other methods were originally proposed in a single-agent setting. Xu et al. (2022) later extended them to the multi-agent case. We believe that the absence of the multi-agent component in Yu et al. does not trivialize their method nor hinder single-agent performance, as they showcase their method also in single-agent exploration. This method still serves as a useful comparison in the single-agent case, as this is a common approach for single-robot path planning.
>
> __Total variation contribution is slightly overclaimed, it is simply the perimeter.__ The key is that TV can be computed locally and continuously, enabling e.g. variational image enhancement, infilling, and segmentation methods (a body of several hundred papers). The insight that these properties also enable its application in continuous RL is, to the best of our knowledge, novel.
>
> __What is the collision frequency?__ For exploration, it varied between once every 100-1000 seconds, which is roughly once every 50-500 meters. For lawn mowing, it varied between once every 150-250 seconds, which is roughly once every 30-50 meters. However, the vast majority of collisions were near-parallel, and we didn’t see any head on collisions. The practical implications are very different between these cases. We will add this analysis to the appendix.

---

> > ### Comment · Reviewer_5fHE · 2023-11-18
> >
> > Unfortunately, the authors have not addressed the bulk of my concerns. It's quite standard for learning for combinatorial methods [1,2] to compare with strong non-learning solvers in problems like TSP and VRP, and here the frontier coverage problem is a minimum length Hamiltonian path problem over a set of frontier points that need to be visited; this problem can be easily reduced to a TSP and fed into a solver. The baselines that the authors used are too fuzzy in my opinion (e.g. potential field), and not principled enough to serve as a strong reference point for the proposed learning-based work. There is little detail in the paper about the implementation and architectures used in the other learning-based baselines, but at least one uses a MLP without CNNs, so it does not seem well-tailored for the problem. I highly encourage the authors to add a TSP baseline with a solver like Concorde.
> >
> > [1] Kwon, Yeong-Dae, et al. "Pomo: Policy optimization with multiple optima for reinforcement learning." Advances in Neural Information Processing Systems 33 (2020): 21188-21198.
> >
> > [2] Hottung, André, Yeong-Dae Kwon, and Kevin Tierney. "Efficient active search for combinatorial optimization problems." arXiv preprint arXiv:2106.05126 (2021).
> >
> > Regarding the TV contribution, it doesn't really matter to the coverage path planning problem whether TV can be computed locally or not, and it's very dubious (and unsupported by the paper) that other image related methods (enhancement, infilling, and segmentation) are relevant to the structure of the problem at hand. To me, it seems that TV merely coincides with the perimeter, which is helpful to minimize for this task. I still do not see TV as some sort of application of computer vision-related insight to this domain. It is my opinion that the authors could be clearer about that, but this is just a minor point.
> >
> > Regarding the collision frequency: I see, that sounds good to me.

---

> > > ### Author Response · Authors · 2023-11-20
> > >
> > > Hopefully the following replies answer the remaining concerns.
> > >
> > > __It's quite standard for learning for combinatorial methods [1, 2] to compare with strong non-learning solvers in problems like TSP and VRP, and here the frontier coverage problem is a minimum length Hamiltonian.__ According to the provided references, TSP seems to assume that all nodes are known beforehand. However, this is not the case in the problem formulation that our method and the compared algorithms address. Sure, a TSP solver could find a path between the current frontier nodes, but it would have to be repeated once new ground is explored and the nodes are updated, which is usually every time step. This is infeasible as the runtime for Concorde seems to be in the order of minutes with 20 nodes. Alternatively, the TSP solver could be run only when a sufficiently large area has been discovered, which would require a strategy for when to run it. To summarize, it is not trivial how a TSP baseline would be formulated in our setting, which would be a contribution in and of itself. Instead, we surveyed the literature to find suitable methods to compare with.
> > >
> > > __The baselines that the authors used are too fuzzy in my opinion (e.g. potential field), and not principled enough to serve as a strong reference point for the proposed learning-based work.__ We refer to the papers of these methods, which contain full details.
> > >
> > > __There is little detail in the paper about the implementation and architectures used in the other learning-based baselines, but at least one uses a MLP without CNNs, so it does not seem well-tailored for the problem.__ (1) Active Neural SLAM uses a CNN. (2) Hu et al. (2020) use a non-learning based method to process the maps and compute the next frontier goal, which to us seems well-tailored. Their RL agent then predicts control signals based on relative goal pose, previous velocity, and lidar data. None of these observations are spatially correlated in 2D, so a 2D CNN would not make sense. One could argue that a 1D convolution could be used for the lidar data, although we believe that this is not what the reviewer was referring to?
> > >
> > > __Regarding the TV.__ Many works exist on perimeter estimation of digital objects [1, 2, 3], which is not the same as the TV in the discrete case. Consider for example the below “images”, which have TV 6.83 / 7.41 / 5.41. Considering both the 4- and 8-neighborhood, the perimeter differs from the TV. Additional problems arise when the objects contain holes.
> > >
> > > 0000\
> > > 0010\
> > > 0110\
> > > 0000
> > >
> > > 0000\
> > > 0110\
> > > 0110\
> > > 0000
> > >
> > > 0000\
> > > 0000\
> > > 0110\
> > > 0000
> > >
> > > [1] Rosenfeld, Azriel. "A note on perimeter and diameter in digital pictures." Information and Control, 1974.
> > >
> > > [2] Klette, Reinhard, Vladimir V. Kovalevsky, and Ben Yip. "Length estimation of digital curves." Vision geometry VIII, 1999.
> > >
> > > [3] Cotsakis, Ryan, Elena Di Bernardino, and Thomas Opitz. "On the perimeter estimation of pixelated excursion sets of two-dimensional anisotropic random fields." Scandinavian Journal of Statistics, 2023.

---

> > > > ### Comment · Reviewer_5fHE · 2023-11-20
> > > >
> > > > Please see Table 1 of this paper [1]. Concorde solves a size 200 TSP in 1.69s (thus size 20 TSP should be trivial), so this could absolutely be run in real-time or with replan once every few steps. Thus, I highly recommend that the authors include a comparison with this type of approach (either for this submission or for future submissions).
> > > >
> > > > [1] Cheng, Hanni, et al. "Select and Optimize: Learning to solve large-scale TSP instances." International Conference on Artificial Intelligence and Statistics. PMLR, 2023.
> > > >
> > > > Regarding the other neural methods, thanks for the clarifications. I will discuss these with other reviewers if necessary to get a better sense of their significance.
> > > >
> > > > Regarding the TV in its application to this work, I would argue that these image processing itself is irrelevant to this domain. What's relevant here is that there is no holes during coverage path planning (since the robot cannot visit disconnected components of the map). I'm merely pointing out that using "TV" in the paper rather than perimeter does not offer additional insights. This is a minor point and I don't believe the authors can change my view on this with additional discussion.

---

> > > > > ### Author Response · Authors · 2023-11-21
> > > > >
> > > > > We were mistaken, the runtime of minutes was over many instances of 20-node problems, 1.69s for 200 nodes is indeed the runtime for one problem. This makes it more feasible. We will implement such a baseline if time permits, although it will not provide an optimal solution due to unknown information. For example, two frontier nodes could be connected over the unexplored region.
> > > > >
> > > > > Holes can still exist in the coverage map in the form of "negative holes", i.e. non-covered regions can be disjoint, which occurs during training. See for example Figure 11 in the supplementary material.

---

> ### Comment · Reviewer_5fHE · 2023-11-21
>
> Thank you. Just a note, TSP solvers typically allow you to specify a distance matrix between every pair of points, so ideally you would try to quantify if this makes a difference from Euclidean distance (on a grid you can get the known shortest paths between any two points with the A* or Bellman Ford algorithm). If Concorde is somehow not a good fit, LKH-3 is another commonly used TSP solver and can be downloaded from http://webhotel4.ruc.dk/~keld/research/LKH-3/. And also I don't necessarily expect the proposed RL method to perform better, but knowing the tradeoff between solution quality and planning time of a TSP solver would significantly help characterize the proposed method.
>
> The negative holes is a fair point, and I agree that "perimeter" is a crude term for cases like this, but I think there should be a better term for the total boundary which gives more insight on a global level than TV, which is a local operation that you apply globally.

---

### Official Review · Reviewer_K43j · 2023-10-30

**Soundness:** 3 good
**Presentation:** 3 good
**Contribution:** 2 fair
**Rating:** 5
**Confidence:** 4

**Summary:**

The study focuses on RL applications for online CPP and investigates several components like action space, input representation, neural networks, and reward functions. Extensive experiments are carried out to evaluate the effectiveness of the method.

**Strengths:**

1. The research topic is interesting.
2. An instance of reinforcement learning applied to the domain of path planning.

**Weaknesses:**

1. The research appears to have limited novelty, with a modest contribution.
2. Most machine learning and classical planning methods can handle the investigated problem. It is hard to find new insights from this study.
3. In terms of multi-scale approaches, prior studies have delved into this issue, notably employing RL in autonomous driving as illustrated in reference [1].

[1] Gu, S., Chen, G., Zhang, L., Hou, J., Hu, Y., & Knoll, A. (2022). Constrained reinforcement learning for vehicle motion planning with topological reachability analysis. Robotics, 11(4), 81.

**Questions:**

1. How might the research tackle challenges related to sim-to-real transfer, especially considering the difficulties of implementing one-shot learning in real-world RL applications?
2. How does the agent architecture in this study differ from architectures used in methods like TRPO?

---

> ### Author Response · Authors · 2023-11-17
> **Replies to reviewer K43j**
>
> We thank the reviewer for their feedback, and provide replies to the concerns below.
>
> __The research has limited novelty, with a modest contribution.__ May we kindly ask the reviewer to be more specific, e.g. by giving a reference that would limit the novelty? As far as we know, previous work has not applied RL to learn continuous control signals end-to-end in CPP. In our interpretation, the strengths outlined by reviewers aapF and 5fHE tend to suggest that they agree. Learning control signals end-to-end in this way comes with unique challenges, which we address in the paper.
>
> __Most machine learning and classical planning methods can handle the investigated problem.__ We agree that many classical and learning-based methods tackle a similar problem. However, none approach it end-to-end with RL as mentioned above. Compared to classical methods and multi-stage frameworks, this approach does not put any constraints on the path space, and allows the model to adapt to specific quirks in the agent dynamics in order to find a more efficient path.
>
> __In terms of multi-scale approaches, prior studies have delved into this issue.__ This is true. We are not suggesting that it is a new concept, but rather, that it is a necessary part in our solution to the challenge of applying RL to large-scale CPP. We will clarify this part in the paper.
>
> __How might the research tackle sim-to-real transfer?__ Good question. As mentioned early on page 4, we believe that our approach can be used with an off-the-shelf SLAM method, removing the need for known pose. In this setting, the agent can learn and adapt to the noise characteristics during training. That our approach works well on noise-free data / data with simulated noise is a necessary condition for future real-world applications. The evaluation of our method on physical robot systems with realistic, noisy sensing will be subject of a future manuscript.
>
> __How does the agent architecture differ from architectures in methods like TRPO?__ The architectures used in TRPO are either feed-forward fully connected networks, similar to our MLP baseline, or convolutional networks for image observations, similar to the map feature branch in our CNN baseline. They are different in regards to our multimodal feature extractor and SGCNN architecture.

---

### Official Review · Reviewer_aapF · 2023-11-01

**Soundness:** 3 good
**Presentation:** 3 good
**Contribution:** 2 fair
**Rating:** 3
**Confidence:** 5

**Summary:**

The paper presents an end-to-end reinforcement learning approach to solve the problem of online coverage path planning. In particular, the paper formulates the problem as a Markov Decision Process, where the action space is continuous (linear and angular velocity), the observation space includes a multi-scale map representation, and the reward function includes a total variation term to incentivize the agent not to leave any unexplored part behind. The evaluation and comparison are performed on a number of different environments and with respect to some classic and learning-based methods. The code is included.

**Strengths:**

- Overall, the paper presents an RL based end-to-end method able to have the agent explore different unseen environments, providing an interesting view on the representation needed to achieve such a task.

- with a standard RL problem formulation, the proposed method includes elements that are not present in the current learning-based methods, such as including continuous action space.

- the presentation overall is clear, with a logical structure and justifications included in the different choices.

**Weaknesses:**

- the paper provides as motivation with respect to classic methods "As these classical approaches cannot adapt to non-modeled environment dynamics, we turn to learning-based approaches", however, also the proposed method does not include any environment dynamics (the paper includes some examples in the introduction, such as damaged sensor or actuator). The paper currently lacks the corresponding gap from the classic methods and does not address the issue mentioned.

- while the paper includes a number of different methods for comparison, there are some methods that are not discussed at all, but that however would be relevant to discuss, in particular some learning-based methods that are predicting the structure of the environments, such as

Caley, Jeffrey A., Nicholas RJ Lawrance, and Geoffrey A. Hollinger. "Deep learning of structured environments for robot search." Autonomous Robots 43 (2019): 1695-1714.

Shrestha, Rakesh, Fei-Peng Tian, Wei Feng, Ping Tan, and Richard Vaughan. "Learned map prediction for enhanced mobile robot exploration." In 2019 International Conference on Robotics and Automation (ICRA), pp. 1197-1204. IEEE, 2019.

Arguably, compared to classic methods, a learning based approach can potentially learn some patterns to choose locations that are promising in terms of new information.

- there are parts that are not fully realistic, in particular the noise that would be present in the partial maps built by the robot, which has the effect of potentially guiding the robot towards areas that do not require exploration. It would be instead good to actually have the map built considering the noise. This can be achieved by using a realistic robot simulator with the perception/navigation stack already existing, e.g., in ROS.

- it is also not clear why different methods are used for comparison in the omnidirectional and non-omnidirectional case, especially for the classic methods. Classic methods typically separate the high level decision and planning to the low-level control, so the same methods can be applied.

- to appreciate the difference between methods, it is important to include also the standard deviation, i.e., in Table I. It would have been good to include coverage over time also for the omnidirectional case, in the appendix.

- the multiscale component plays a role given the fixed w and h for the map, however, it is not clear the difference in performance between a multi-scale approach vs if those w and h would be high enough to capture the environment at enough fine resolution. It would be interesting to see such a difference. In addition, there might be environments where one scale is enough, while other complex environments require more than one scale. It would be interesting to discuss such a difference in different environments, thus hinting an adaptive scale, and how in practice an adaptive scale can be included, as the training will happen with specific scales.

- a comment should be included about the impact of normalizing the distance measurements with respect to the maximum range, in particular about the generalization of the learned policy on a robot with different sensor range.

- it is good to provide an intuition on how to set the parameters

Just some minor presentation comments to fix:
- "in (1)" -> "in Eq. (1)"
- it can make sense to change CPP to exploration, given that in the end the coverage, for example in the lawn mowing task, has quite some difference in performance.

**Questions:**

- what is the standard deviation of the results in Table I? in other words, are the differences statistically significant?

- please include results with the same methods for both omnidirectional and non-omnidirectional case.

---

> ### Author Response · Authors · 2023-11-14
> **Reply to first two points**
>
> We thank the reviewer for their valuable feedback on how to improve the quality of our paper. We address the first two points. Additional answers will follow shortly.
>
> __The proposed method does not include any environment dynamics.__ The ability to adapt to non-modeled environment dynamics was mainly used as a motivation for choosing an RL-based approach. Indeed, [1] shows that in grid environments, RL can be robust to dynamic environments and outperform classical methods. We missed to include this reference and will update the paper accordingly. Due to the findings in [1], we focused on the question whether an RL-based approach can learn patterns to choose locations that are promising in terms of new information in a continuous setting at all, before moving to dynamic environments.
>
> [1] Saha et al. “Online area covering robot in unknown dynamic environments”, ICARA, 2021.
>
> __Some methods are not discussed at all.__ There are many works on CPP that share a connection to our work. However, to include all of them would not fit within 9 pages. We had to make a selection and choose the most related ones. We did not include Caley et al. (2019) as it tackles a slightly different problem, robot search (i.e. find the exit) instead of covering the area completely. Shrestha et al. (2019) presents and interesting approach for the same task and we thank the reviewer for this reference, we will include it in the related work.

---

> ### Author Response · Authors · 2023-11-17
> **Additional replies**
>
> Here follows additional replies.
>
> __The noise in the built maps are not fully realistic.__ As mentioned early on page 4, we believe that our approach can be used with an off-the-shelf SLAM method, removing the need for known pose. In this setting, the agent can learn and adapt to the noise characteristics during training. That our approach works well on noise-free data / data with simulated noise is a necessary condition for future real-world applications. The evaluation of our method on physical robot systems with realistic, noisy sensing will be subject of a future manuscript.
>
> __It would be good to build the maps considering the noise using a realistic robot simulator, e.g. in ROS.__ We agree that this would be highly interesting, although ROS is not necessarily fully realistic as it does not have a full notion of continuous time. Evaluations on real robots would be preferable to any simulation.
>
> __Why use different methods to compare omnidirectional/non-omnidirectional exploration?__ We simply compared the methods on the task setup described in their respective code implementations. We will provide links to the code repositories used in the appendix.
>
> __The standard deviation should be included, e.g. in Table 1.__ We will do our best to do multiple runs for the standard deviation, although we can not promise that they will finish during the discussion period.
>
> __It would be good to include coverage over time in the omnidirectional case in the appendix.__ Good suggestion. Although, the T90 and T99 metrics used in Explore-Bench convey information part of such a plot.
>
> __How does a multi-scale map compare to a single map with high enough w and h?__ In large environments, using a single map is not feasible as the computational cost is O(n^2). For example, with our current setup with 4 maps that span an area of size 76.8x76.8 m2, the training step time is 40 ms. With a single map spanning only 19.2x19.2 m2 using the same pixel resolution as the finest scale, the training step time is 2500 ms. This would increase the total training time by roughly two orders of magnitude. Thus, a multi-scale map is actually necessary for large-scale environments. We will clarify this part in the paper.
>
> __Could an adaptive scale be included?__ As you mention, the training needs to be done with specific scales. We believe the most practical way is to simply train with sufficiently many scales to cover the maximum size for a particular use case. If the model is deployed in a small area, the larger scales would not contain any frontier points in the far distance, but the agent can still cover the area by utilizing the smaller scales. Increasing the represented area by adding more scales is fairly cheap, as the computational cost is O(log(n)). We will add such a discussion to the appendix.
>
> __The impact of normalizing the distance measurements should be discussed.__ Good point. With this normalization, the training is tied to a specific sensor range. However, the map representation is more flexible, and does in theory also contain the information of the raw lidar data. So excluding the sensor feature extractor in Figure 4 would in theory generalize to any sensor setup. We will add this discussion to the appendix.
>
> __It is good to provide an intuition on how to set the parameters.__ Good point. We will describe the parameters in more detail and provide an intuition on how to set them in the appendix.

---

> > ### Comment · Reviewer_aapF · 2023-11-20
> >
> > Thanks for the response to this and others' reviews. The additional clarifications that are planned to be added in the paper or the supplementary material improves the overall quality of the paper.
> >
> > While the advantage of RL-based methods over classic methods for dynamic environments is shown in at least another paper, it is important to show that the proposed method would still work in case of dynamic environments. Note that the general idea of using the structure of the environment to learn promising locations is not new. In fact, Caley et al. exploited the same concept in the context of search, although the underlying routine is based on exploration towards frontiers.
> >
> > Also, about the noise, simulators in ROS, such as Gazebo, are already a step towards real robots and in robotics they are used. In practice, while the simulator has a simulation timestep, for testing purpose it can be used for continuous cases. In fact, even real robots have sensor measurements and control commands at a frequency.
> > In addition, the 2D occupancy grid map produced with the laser sensor can be quite noisy for example with free space that might appear behind obstacles.
> > It is important thus to test the method at least in the simulator and clearly even better on real ground robots.
> > Note that ROS is a middleware that in its core functionality provides ways for processes to talk to each other with a publish/subscribe paradigm and standardized messages. In this way, the code written and tested within a simulator can work out of the box with a real robot.

---

> > > ### Author Response · Authors · 2023-11-22
> > >
> > > We agree that it is interesting to evaluate also on dynamic environments, although this was beyond the scope of this paper, as stated in Section 3.1. The results indicate that our approach shows promise for real-world scenarios, which is why we are currently working towards a ROS implementation for a real robot system, where we aim to include dynamic obstacles.

---

### Official Review · Reviewer_HYy8 · 2023-11-08

**Soundness:** 3 good
**Presentation:** 3 good
**Contribution:** 2 fair
**Rating:** 6
**Confidence:** 5

**Summary:**

This paper presents a reinforcement learning approach for online coverage path planning in unknown environments. The authors use a continuous action space for the agent in their formulation, with the model directly predicting the low level control signals. When compared to prior work, the main contribution seems to be the choice of observation representations and the reward function. Observation data is represented through multi-scale egocentric maps to enable long-term planning while maintaining scalable input representation. Additionally, the introduction of frontier maps helps retain information about uncovered spaces in larger-scale maps. The authors propose a reward function with a total variation term to reduce small uncovered areas within the environment. The approach is tested on tasks such as exploration and lawn mowing, showing improved performance over existing reinforcement learning and classical path planning methods.

**Strengths:**

- The authors combine a lot of well-established ideas and apply them to the problem of interest - multi-scale egocentric maps, frontier encoding, along with the intuitive total variation term in the reward. The impact of these elements is clearly demonstrated.
- Experiments across different 2D environments demonstrate general applicability of the learned policy and the performance seems to exceed the baselines that include classical methods and learning-based methods. The analysis is thorough with ablation studies of different components including observation representations, rewards, and architectures.
- The paper is well-written and organized, and clearly motivates different concepts before getting to the solution approach. The notation is consistent and clear throughout the paper and the figures are informative (with one exception, see weaknesses). Great job with the writing!

**Weaknesses:**

- The details of the agent architecture and the reasoning behind the design choices isn’t completely clear - is there a reason for selecting SAC over other RL algorithms? What was the motivation behind the MLP architecture as a baseline when it’s not the best choice of the input representations at hand? Figure 4 is a bit confusing too - I thought multi-scale frontier maps M_f are a par t of the observation too but the illustration in Fig 4 don’t seem to mention that. Also, it wasn’t super clear at first glance what x3 or x4 meant in the architecture.
- I am not sure how practical the proposed approach is in real-world deployments - at the end of the day, the goal is to be able to deploy learned policies on real-systems. However, perfect observations for building global coverage/obstacle map and noise-free position/pose information are fairly strong assumptions that might limit practical utility. I do acknowledge the experiments with added Gaussian noise, but the scale of the noise isn’t completely clear and it seems like noise was only added to position information. I would have loved to see some real-world experiments or experiments on high-fidelity simulators with some discussion on inference speed/real-time performance.
- While I understand the work emphasizes on coverage path planning, I would have expected some discussion/experiments on the extent of collisions too - collisions matter in real-world problems, and unlike classical methods, the extent of collisions in a learned policy would at the end of the day depend on the relative weight of the collision-avoidance reward. It would be interesting to see the trade-off in the performance with more emphasis on collision avoidance when compared to classical approaches (to be clear, I am not recommending new experiments here - but it would be great if authors could discuss this aspect/share more details if they have the necessary information in the logs of existing runs).

**Questions:**

In addition to the questions in the weaknesses section, I have the following questions/suggestions:

- The word “environment dynamics” the the text is confusing - I understand what the authors wanted to convey but at the end of the day it’s agent dynamics.
- There are some issues with the notation in the POMDP definition. The state is Markovian by definition, and probability of transitioning to s_t should only depend on the previous state and action. If the problem formulation requires violating this, mention it explicitly.
- Did you experiment with any other reward formulations besides coverage area and total variation?
- How does the performance scale to much larger environments? At some point the multi-scale representation may need higher resolution or more scales, right?
- What’s the inference time like say on a standard laptop?

---

> ### Author Response · Authors · 2023-11-17
> **Replies to reviewer HYy8**
>
> We thank the reviewer for their valuable suggestions on improving the clarity, motivation and analysis. We are also encouraged by the positive feedback. We address the concerns below.
>
> __What is the reason for choosing SAC over other RL methods?__ In our understanding, SAC is the standard method for continuous RL, and has been used successfully in robot navigation [1]. It also has high sample efficiency, which is important for future real-world experiments.
>
> [1] de Jesus et al. "Soft Actor-Critic for Navigation of Mobile Robots", Journal of Intelligent and Robotic Systems, 2021.
>
> __What is the motivation for the MLP baseline?__ The purpose of the MLP baseline is to evaluate the effect of the inductive biases in the CNN/SGCNN architectures. Note that the MLP architecture even has more parameters, 3.2M compared to 0.8M for the CNNs, due to the initial fully connected layer. This tells us that the inductive biases are indeed necessary.
>
> __Figure 4 is missing the frontier maps.__ Yes, we realized, we will update the figure accordingly.
>
> __It is unclear what x3 and x4 mean in Figure 4.__ We can see that, we will update the caption explaining that they refer to the number of conv and fully connected layers.
>
> __Perfect observations and noise-free pose might limit practical utility.__ As mentioned early on page 4, we believe that our approach can be used with an off-the-shelf SLAM method, removing the need for known pose. In this setting, the agent can learn and adapt to the noise characteristics during training. That our approach works well on noise-free data / data with simulated noise (see below) is a necessary condition for future real-world applications. The evaluation of our method on physical robot systems with realistic, noisy sensing will be subject of a future manuscript.
>
> __The scale of the noise isn’t completely clear.__ We measured the average position error for the SLAM method used in the CPP methods under comparison, and applied a similar noise to our method to get a fair comparison.
>
> __Experiments on high-fidelity simulators.__ We agree that this would be highly interesting, although evaluations on real robots are preferable to any simulation.
>
> __What is the inference time?__ The model inference time is 5 ms on a laptop (i5-520M CPU, no GPU), and 2 ms on a high-performance cluster node (6226R CPU, T4 GPU). As our network is fairly lightweight it can run in real time. The time for updating the maps and constructing the multi-scale map representation etc. is 57/9 ms for exploration/mowing on the laptop, and 19/3 ms for exploration/mowing on the cluster node. All map updates are performed locally, resulting in high scalability. We will add this analysis to the appendix.
>
> __What is the collision frequency?__ For exploration, it varied between once every 100-1000 seconds, which is roughly once every 50-500 meters. For lawn mowing, it varied between once every 150-250 seconds, which is roughly once every 30-50 meters. However, the vast majority of collisions were near-parallel, and we didn’t see any head on collisions. The practical implications are very different between these cases. We will add this analysis to the appendix.
>
> __The word “environment dynamics” is confusing, it should be “agent dynamics”.__ By “environment dynamics” we meant to convey aspects that do not only depend on the properties of the agent, such as wheel slip or moving obstacles.
>
> __In the POMDP definition, the probability of transitioning to s_t should only depend on the previous state and action.__ This is true, the state should not depend on all previous states and actions leading up to the new state (which we do not violate), we will update this.
>
> __Did you experiment with other reward formulations?__ Yes, we did. We tried a positive goal reward for reaching the goal coverage, and a negative truncation reward, but these gave no effect. We will include them in the appendix.
>
> __How does the performance scale to much larger environments? More maps or higher resolution is required, right?__ Yes, more maps or higher resolution would be needed for much larger environments. More maps would be most feasible, as the computational cost is O(log(n)). For example, our current setup with 4 maps spans an area of size 76.8x76.8 m2 with 4096 pixels. Increasing the number of maps to 6 would increase the computational cost by 50% while spanning more than 1x1 km2 in 6144 pixels.

---

### Comment · Area_Chair_hpe2 · 2023-11-20
**Author-Reviewer Discussion Period Ending November 22**

Hi,

Thank you for your help with the review process!

There are only two days remaining for the author-reviewer discussion (November 22nd). If you haven't already done so (thanks Reviewer 5fHE), please read through the authors' response to your review and comment on the extent to which it addresses your questions/concerns.

Best,\
AC

---

### Author Response · Authors · 2023-11-22
**Paper revision**

Thank you to all the reviewers for the constructive feedback and discussions, with helpful suggestions for improvements to our paper. We have now revised the paper with the promised modifications.

---

### Meta-Review · Area_Chair_hpe2 · 2023-12-11

**Metareview:**

The paper describes a deep reinforcement learning (RL) approach to coverage path planning problems (e.g., exploration and lawn mowing) in a priori unknown environments. The method reasons over a multi-scale egocentric map of the explored environment and its frontier regions to facilitate efficient, long-term planning. The paper proposes to augment the reward with a total variation term that encourages the agent to not leave any regions of the environment unexplored. Experiments on simulated exploration and lawn mowing problems demonstrate that the proposed method outperforms classical and learning-based baselines.

One of the paper's primary contributions is the demonstration that existing concepts including the use of multi-scale egocentric maps as the observation representation with frontier regions, and total variation-based reward densification can be combined to perform coverage path planning. As three of the reviewers point out, the paper is well organized and well written. However, the reviewers identify a few key limitations with the current version of the paper that make the significance of its contributions unclear. Reviewers 5fME and HYy8 question the sufficiency of the baselines that the paper evaluates against. Reviewer 5fHE emphasizes that the framework should be compared to a non-learning-based solver like TSP or another combinatorial optimization-based baseline. They also find it difficult to ascertain the adequacy of the existing baselines, many of which are designed for a multi-agent setting, in part due to the lack of details regarding their implementations. Meanwhile, two reviewers (HYy8 and aapF) raise concerns about the practicality of the method due to the assumption that the observations are noise-free (i.e., they involve perfect maps) and that the pose estimates are similarly noise-free. The paper does offer experiments that include added Gaussian noise, though Reviewer HYy8 comments that the extent to which it provides a reasonable approximation for real-world noise is unclear. While there was a healthy discussion between the authors and a subset of the reviewers, several of primary concerns that the reviewers initially raised were not adequately resolved. The authors are encouraged to address these concerns in any future version of the paper.

Note: The AC acknowledges that Reviewer K43j did not reply to the authors' response, despite several attempts by the AC to encourage their involvement. As a result, the AC placed less weight on their review when making a recommendation.

**Justification For Why Not Higher Score:**

With the exception of Reviewer K43j, the reviewers read and replied to the authors' response to their initial reviews, with Reviewer HYy8 actively discussing the paper with the authors. At the conclusion of the author-reviewer discussion, the three reviewers felt that their primary concerns remained, which the AC agrees with.

**Justification For Why Not Lower Score:**

N/A

---

### Decision · Program_Chairs · 2024-01-16

Reject